# Evaluation of Morphological and Structural Skin Alterations on Diabetic Subjects by Biophysical and Imaging Techniques

**DOI:** 10.3390/life13020579

**Published:** 2023-02-18

**Authors:** Veronica Rego Moraes, Maisa Oliveira Melo, Patrícia M. B. G. Maia Campos

**Affiliations:** School of Pharmaceutical Sciences of Ribeirão Preto, University of São Paulo, Ribeirão Preto 14040-903, SP, Brazil

**Keywords:** skin barrier function, collagen, diabetes, reflectance confocal microscopy

## Abstract

Diabetes causes increased production of advanced glycation end products (AGEs), which may lead to irreversible damage to collagen fibers, and early and more accentuated signs of skin aging. Thus, the objective of this study was to evaluate diabetic skin’s mechanical and morphological characteristics and compare these to healthy skin. Twenty-eight female participants aged between 39 and 55 years were enrolled: half had type 2 diabetes, and the others were healthy. Wrinkles, transepidermal water loss (TEWL), stratum corneum water content, skin color, elasticity, morphological and structural characteristics of epidermis and dermis echogenicity were evaluated using biophysical and skin imaging techniques. Higher TEWL values were observed in participants with diabetes, who also showed lower skin elasticity and wrinkles with greater volume, area, and depth. In addition, the Reflectance Confocal Microscopy (RCM) imaging analysis showed that all participants with diabetes presented polycyclic papillae and deformed and amorphous collagen fibers. The obtained data showed significant differences between healthy and diabetic skin and could help develop more specific topical treatments to improve the treatment of skin conditions in people with diabetes. Finally, RCM is an advanced imaging technique that allows for a more profound analysis of diabetic skin, which could assist in the evaluation of dermocosmetic treatments to improve the skin alterations caused by this disease.

## 1. Introduction

Healthy skin is associated with the image of skin with a better, clearer, and less sensitive appearance. When skin is healthy, it has several protective functions against external microorganisms and exposure to ultraviolet light, among other threats, in action [1]. On the other hand, skin aging causes structural changes that result in thinning and impaired epidermal barrier recovery. Like other organs, skin aging is known for the progressive loss of functionality and regenerative potential [2]. Aging occurs through several factors, both intrinsic and extrinsic. Intrinsic factors result from biological processes, such as oxidation and glycation reactions. Extrinsic factors, on the other hand, are correlated with several variants such as exposure to UV, smoking, sedentary lifestyle and diet. Wrinkles, fine lines, sagging, difficulty healing, and pigmentation changes are some clinical signs of skin aging. At a molecular level, many alterations can be observed; however, collagen is the one that suffers more damage, with the passage of time causing its fragmentation, which results in the loss of the mechanical properties of the skin [2].

The glycation process is derived from a non-enzymatic aminocarbonyl reaction structured from interactions between reduced sugars and proteins [3]. The glycation reaction produces a group of chemical moieties known as advanced glycated end products (AGEs). AGEs are primarily responsible for more serious damage in patients with Type 2 diabetes.

AGEs can be found in several types of tissue during aging and in diabetes patients, such as in the skeletal and smooth vascular muscles, basement membranes, and articular collagen, among others [4]. AGE concentration increases in the body during the aging process, which may cause structural collagen loss and compromise the integrity of the tissue [5].

An excessive accumulation can be caused by pathologic diseases like diabetes mellitus and cardiovascular diseases. Human aging can also lead to an accumulation of AGEs and cause a stiffening of tissues like tendons, joints, bones, arteries, and skin, which can lead to several health problems such as neurodegeneration and diabetic neuropathy [6,7]. When it comes to the skin, the most serious damage caused is that collagen fibers are modified due to Maillard reactions. This process begins with the Schiff base reaction, which condenses the carbonyl group of a reduced sugar with an amine group derived from an amino acid, after which this base undergoes structural rearrangements that result in a more stable product, called the initial product of the reaction of Maillard and Amadori products. These compounds may interact irreversibly with the amino acid and transform into advanced glycation end products [8].

These physiological changes in fibers are being studied, especially in people with diabetes mellitus as Endogenous AGEs are formed more rapidly in people who have this disease [8]. This happens due to the high concentration of circulating sugar in the blood, which generates greater degradation in the collagen and, consequently, more perceptible and accentuated signs of aging [4].

Considering that diabetes mellitus disease afflicts more than 8.5% of the world’s adult population and is associated with a death toll of over 3.7 million per year, proposals for anti-AGE products are important. The pharmaceutical and cosmetics industries are developing new compounds and products with anti-aging characteristics for this type of skin [4].

Furthermore, the skin of a person with diabetes may present an impaired skin barrier because of their dysfunctional metabolism, which can cause a decrease in the hydration of the stratum corneum and increased transepidermal water loss [9,10].

Cutaneous problems are commonly found in people with type 2 diabetes, such as skin tags and ulcers. However, little is known about the condition of the skin barrier of people with type 2 diabetes. Studies show that there are functional changes in the skin of both rats and humans with type 2 diabetes. The change in the hydration of the stratum corneum can occur in several ways, among them, the decrease in aquaporin 3 and the lower proliferation of keratinocytes that is related to the dry skin of people with diabetes [11].

Among the biophysical noninvasive techniques used in the clinical efficacy area, we highlight the use of devices to study skin color, elasticity, stratum corneum water content, and skin barrier function integrity, as these parameters are essential to maintaining healthy skin [12]. Biophysics and skin imaging noninvasive techniques can analyze the clinical effects of these proposed products. Through skin imaging techniques and innovative methods, it is also possible to study the dermis’ echogenic characteristics through a high-frequency ultrasound.

Another important technique that has gained prominence in in vivo skin characterization and efficacy studies is Reflectance Confocal Microscopy (RCM). RCM is a non-invasive technique that evaluates the morphological and structural characteristics of the skin and can be applied in the evaluation of topical formulation efficacy, as it makes it possible to observe the skin’s real-time condition and offers epidermal parameters that can be used in both skin characterization and treatment follow-up [13].

This way, the characterization of this type of skin by advanced skin imaging techniques will help discern the differences between these patients and those with normal skin and how the glycation process interferes with the aging process, guiding the development of new, effective treatments for both populations.

In addition, the role of AGEs in the skin aging process is a research target for many scientists [5]. However, more studies and specialized products are still needed to better understand its mechanisms and how it differs from a healthy skin aging process.

Finally, the study of this form of aging can lead to the development of specific cosmetics for people with diabetes since they have peculiarities in their skin barrier. In addition, this research can help to minimize the chances of having an adverse reaction [14], at the same time helping to create loyalty of the consumer to the cosmetic industry that developed it.

### Objective

The aim of this study was to evaluate the characteristics of aging in diabetic skin and compare them to healthy skin using biophysical and skin imaging techniques.

## 2. Materials and Methods

### 2.1. Study Design

This study was approved by the ethics committee of the School of Pharmaceutical Sciences (CEP/FCFRP n°. 451–CAAE n° 73911117.6.0000.5403), and all subjects provided written informed consent.

### 2.2. Participant’s Trials

The study was separated into two screenings: in the first stage, 28 female participants aged between 39 and 55 years (±49.07) with Fitzpatrick skin phototypes II-III were recruited and separated into two groups. Group 1 consisted of 14 healthy participants and Group 2 of 14 participants with diabetes. The second stage included 12 participants (6 healthy and 6 diabetics), picked out randomly from the first stage, who had their skin analyzed by Reflectance Confocal Microscopy (RCM). After recruitment, an anamnesis was done to understand their health habits, with questions about eating habits, water ingestion, sunscreen use, and cosmetic allergies. The participants with diabetes should have been diagnosed for at least 5 years, and were also asked when their treatment started. The non-inclusion factors were if the participant was a smoker, pregnant, or with dermatological diseases. The experiments were carried out at the School of Pharmaceutical Sciences of Ribeirão Preto, University of São Paulo, Ribeirão Preto, São Paulo, Brazil (21°100 S, 47°480 W) in a room set at 21 ± 2 °C and 45–55% relative humidity. The participants were acclimatized in the room for at least 15 min before taking measurements to allow their skin to fully adapt.

### 2.3. Instrumentation

Biophysical and skin imaging techniques were used to perform the objective measurements. The participants were asked not to use any cosmetics for 24 h and to avoid washing the skin for 2 h before measurements were taken.

The transepidermal water loss (TEWL) (g cm^−2^ h^−1^) was evaluated using the Tewameter^®^ TM 300 equipment (Courage & Khazaka, Cologne, Germany), based on the diffusion principle described by Adolf Fick. The values were given in g m ^−2^ h.

The stratum corneum water content (hydration) was determined using a non-invasive skin capacitance meter Corneometer^®^ CM 825 (Courage & Khazaka, Cologne, Germany), which measures the capacitance that is entirely dependent on the water content in the skin. Different capacitance changes are converted into a digitally measured value (arbitrary units) that is proportional to the skin humidity, and results are given in arbitrary units (AU) which are estimated to correspond to 1 AU in 0.2–0.9 mg of water per gram of the stratum corneum [15].

Colorimetry was tested using the Colorimeter CL 400 device (Courage & Khazaka, Cologne, Germany), which quantifies the reflectance of the standard monochromatic light source on the skin. This study evaluated the *b parameter, where *b represents the yellow/blue coordinate [16].

The Visioface^®^, a digital photography imaging system (Courage & Khazaka, Cologne, Germany) that obtains high-resolution images to evaluate facial skin, was used to analyze the presence of wrinkles and their characteristics. It consists of a cabin attached to a high-resolution digital camera (10 megapixels) and 200 white LEDs and is connected to research software that enables the evaluation of visible spots, pores, wrinkles, and color differences in the target area, which is selected manually. Additionally, the system analyses the area, volume, and depth of the facial wrinkles.

The Cutometer SEM 575 (Courage Khazaka, Cologne, Germany) was used to evaluate skin viscoelasticity. Measurements were taken under the following conditions: aperture, 2 mm; time of application and relaxation, 2 s; and pressure, 450 Mb. The parameters analyzed through the instrument software were: R0—the final distension of the first curve (Uf); R2—the overall elasticity of the skin, including creep and creep recovery (Ua/Uf); R5—the net elasticity (Ur/Ue); R6—the viscoelastic ratio (Uv/Ue); and R7—the ratio of elastic recovery to the total deformation (Ur/Uf) [17].

The dermis thickness and echogenicity were evaluated using a 20 MHz ultrasound Dermascan C^®^ (Cortex, Hadsund, Denmark) containing a transducer focus used for the attainment of two-dimensional transverse images, which are represented in the B-mode software. The ultrasonic wave (speed of 1580 m s^−1^) is partially reflected by the skin structure, giving rise to echoes of different amplitudes. To calculate the echogenicity, the number of pixels with low echogenicity was measured by means of the image analysis software and compared to the total number of pixels. The echogenicity ratio was calculated as the ratio of the number of low-echogenic pixels to the number of total echogenic pixels (LEP/TEP).

The evaluation of epidermal thickness and cellular and tissue characteristics of the epidermis was performed by Vivascope^®^ 1500 Reflectance Confocal Microscopy (RCM) (Lucid, Newark, CA, USA), which allows skin evaluation with a close to histological resolution. Microscopic images were produced in triplicate using the “Vivastack” mode, consisting of multiple confocal images of successive depths taken within a given tissue region. From the skin surface, 1.5 µm stacks were obtained up to a depth of 37.5 µm. Then, 3 µm stacks were obtained up to a depth of 112.5 µm, and 4.5 µm stacks were obtained until a depth of 132.50 µm was reached.

The images obtained by RCM were analyzed according to the recommendations of Longo et al., 2013 [18] and Manfredini et al., 2013 [19], observing the presence of polycyclic papillae and a qualitative 0–3 score of the collagen found in the images, separated into the four types of fibers: thin reticulated collagen, coarse collagen structures, huddled collagen and curled bright structures.

All measurements were performed in the periorbital and nasolabial regions of the face.

### 2.4. Statistical Analysis

Statistical analysis was performed using Graphpad Prism 5. To determine significant differences in skin properties for regions between populations, Student’s *t* test was used for parametric data, and the Mann–Whitney U test was used for non-parametric data. A *p* value < 0.05 was considered significant.

## 3. Results

### 3.1. Biophysical Noninvasive Techniques

#### 3.1.1. Transepidermal Water Loss and Stratum Corneum Water Content

In this study, TEWL values in the periorbital region of the face were significantly higher in the skin of participants with diabetes when compared with healthy skin participants (Figure 1). However, in the evaluation of stratum corneum water content (Figure 2), no significant differences were observed between the healthy and diabetic groups.

#### 3.1.2. Skin Coloration

Significant modification of the skin coloration in the b* values was observed between the study groups. It was possible to observe a more yellowish coloration (Figure 3) in the periorbital region of the participants with diabetes when compared with the healthy group. However, in the nasolabial region this difference was not significant between the groups.

#### 3.1.3. Elasticity of the Skin

In the evaluation of skin viscoelasticity properties as quantified by the R5 (Ue/Ur) parameter, a significant increase in skin net elasticity was observed in the healthy participants compared to the diabetic group (Figure 4). However, no significant difference between R2 (Ua/Uf), R6 (Uv/Ue), and R7 (Ur/Uf) parameters was noted in either studied regions of the face.

### 3.2. Skin Imaging Noninvasive Techniques

#### 3.2.1. Evaluation of Wrinkles in the Skin

In the evaluation of facial skin using high-resolution images, wrinkles had a more extensive area in the diabetic skin (Figure 5) in both the periorbital and nasolabial regions. In the periorbital region, however, it was possible to verify a statistically significant difference between the volume, area, and depth of wrinkles, showing that diabetic skin has larger and deeper wrinkles when compared to healthy skin.

#### 3.2.2. Dermis Echogenicity

Figure 6 shows that the participants with diabetes presented significantly lower echogenic pixels in the periorbital region when compared to healthy skin. The echogenicity ratio (Figure 7) shows that Group 2 also presented lower echogenicity by having a higher echogenicity ratio.

#### 3.2.3. Reflectance Confocal Microscopy

Confocal microscopy is a technique for the microscopic observation of epidermis and dermis aspects. In the morphological analysis of the dermis by RCM imaging analysis (Figure 8 and Figure 9), it was possible to observe multiple polycyclic papillae—which are often found in people who have more pronounced skin aging as they have lost their rounded appearance and defined shapes, becoming dysmorphic agglomerations—and amorphous and thicker collagen fibers in the skin of participants with diabetes.

Furthermore, in Group 2, three participants showed curled collagen fibers, which may suggest the presence of solar elastosis [20]. Skin aging affects collagen. There is a change in the composition of the dermis structure, where there is a decrease in the percentage of reticulated and fine collagen fibers, and an increase in the concentration of dysfunctional, coarse collagen fibers that impair the elasticity of the skin [20]. The other three participants of the diabetic group showed huddled and amorphous collagen which is characterized by a coarser structure and which does not favor dermal support. On the other hand, only one healthy participant showed huddled collagen, while the others did not show any significant changes in their collagen fibers. Thus, diabetes can provoke alteration in the morphological characteristics of dermis with the appearance of thinner stratum corneum, more dysfunctional collagens and low skin cell turnover.

## 4. Discussion

Over the years, the prevalence of diabetes mellitus worldwide has been increasing by high percentages and will continue to expand to include approximately 285 million adults by 2025 [21]. Beyond the known effects of diabetes on the human body, the consequences of this disease on the skin still need to be further investigated if they are to be treated appropriately. In this context, we highlight the increasing interest by the pharmaceutical and cosmetic industries in the glycation process and how it influences skin aging, being a promising target for use in new compounds and products.

This study did not observe any significantly different results in the superficial skin hydration between groups, which is analyzed by the stratum corneum water content parameter [22]. This may be related to the disease itself because, during the anamnesis of the participants with diabetes, it was observed that they ingested a larger amount of water when compared to the healthy group.

The relationship between the superficial hydration of diabetes patients is still poorly understood. Sakai et al. have observed that stratum corneum water content is usually lower in diabetic rats than in healthy animals, but the patient habits could change this scenario in the human skin [23]. On the other hand, Seiraifi et al. did not find any significant difference in the stratum corneum water content between diabetic and healthy patients. However, they suggest that lower hydration may be related to the hyperglycemia that occurs in the disease and may be even more evident in an older group [22].

TEWL is considered a significant parameter in the evaluation of the mechanism and effects of the glycation process as it measures the skin barrier function integrity [24]. In our study, higher TEWL values were observed in the diabetic skin, suggesting that this skin barrier function was more damaged than that of the healthy skin, even in the same age group.

However, Lai et al. evaluated skin hydration in diabetic and healthy patients at six different body sites in both groups and concluded that there was no difference in TEWL and skin hydration between both groups [10]. This suggests that the TEWL can change according to the studied skin region. Furthermore, other studies have also suggested that patients with diabetes tend to show normal hydration, with decreased sebaceous gland activity and skin elasticity but without alteration of the skin barrier function [22,25].

The study of TEWL is also useful in observing whether the active substance is effective in the treatment and maintenance of the skin barrier; when the skin barrier is impaired, higher TEWL values are observed [26].

In diabetic patients, yellow skin and nails have been commonly reported as cutaneous manifestations. The alteration in skin color can mainly be observed in the palms and soles, bone prominences, facial folds, and rims of ears [21]. The yellow color parameter of the diabetic participant’s skin was significantly higher than that of the healthy group since the yellow coloration is correlated with end-product glycosylation, which has a yellowish color [22]. This can be explained since several highly reactive carbonyl compounds are formed and condense with protein amino groups to form yellow-brown fluorescent protein [22].

In addition, collagen, along with other molecules such as elastin, has the main function of providing mechanical support to the extracellular matrix. Aging causes an increase in the degradation of these collagen fibers, resulting in structural and functional changes in the dermis, such as loss of elasticity, a decrease in epidermal thickness, alterations in the appearance of wrinkles, and reduced ability to retain moisture [27,28].

The data obtained in this study corroborated data reported in the literature, as the diabetic skin was less elastic than that of the healthy group. This type of alteration has already been reported in the diabetic population, where altered skin elasticity was detected [29]. This can be related to the AGEs cross-linking with the collagen, its becoming unfunctional and the other results obtained from this same region, where a lower elasticity and a greater presence of wrinkles were noted.

Lower values in the dermis echogenicity can be correlated with an aged dermis and loss in the dermal structure, including collagen fibers [13]. This restructuring is crucial for the tissue repair and organization of the extracellular matrix. Vimentin, a type III intermediate filament protein that is expressed in mesenchymal cells, is essential for various cell functions such as cell motility and wound healing. It is the principal target for AGE modification by N-(carboxymethyl)lysine (CML). The vimentin glycation caused by CML originates a modification structural of the filament, which accelerates the process of aging [7].

Some studies demonstrate that this modification in vimentin causes alterations such as a loss of fibroblast contractility, which could be a factor that accelerates the aging process. This reduced contractility results in the reduction in the ability to reorganize collagen fibers and rearrange the extracellular matrix, which is fundamental for tissue development and repair [20,30].

Finally, in the RCM analyses, we observed differences in aging signs between the group of participants with diabetes and the healthy group. A significantly higher dermal papillae depth was observed in the diabetic participants when compared to the healthy group. All diabetic participants also presented polycyclic papillae, which are elongated and partially anastomosing structures in a ring-like form. Considering that healthy skin papillae are in the form of rings with very well-outlined contours and similar sizes, these alterations lead to a possible morphological and structural difference between both groups, as polycyclic papillae are featured in cases of photoaging and solar lentigo [31].

Although not significant, which was probably due to the individual variations of the participants, the diabetic skin presented a thinner epidermis and stratum corneum. These results are in accordance with the literature on intrinsic aging parameters, which can be attributed to a lower rate of the cell renewal process and alterations in the morphology of skin keratinocytes. In addition, the RCM images showed some differences in keratinocyte morphology in the diabetic group, which corroborates the results of Longo et al., 2013, who observed a disorder related to aging in keratinocytes metabolism leading to loss of polarity and disorderly maturation.

It is possible to observe the prevalence of thin reticulated fibers in young subjects. With the aging process, these kinds of fibers decrease, and other types of collagen can be observed as thick, coarse fibers and amorphous huddles [22,26]. In the qualitative analysis comparison between the groups, the diabetic skin group was observed to have a prevalence of shrunken, amorphous huddles collagen [31], which is more evident in aged skin, even though the groups have the same age. In healthy skin, more integral collagen fibers that are thin and reflective are present similar to young skin [18].

In summary, the application of noninvasive biophysical and skin imaging techniques for the characterization of diabetic skin is very important to support the development of innovative active ingredients and effective products to act against the alterations that are present in this specific type of skin, filling the gaps in the current literature on the features of diabetic skin.

## 5. Conclusions

According to the obtained data, through the use of noninvasive biophysical and skin imaging techniques, it is possible to observe significant differences between healthy and diabetic skin. This can help in the development of more specific topical treatments to improve skin barrier function, elasticity, and dermis density in diabetic patients.

The application of these techniques, and mainly RCM, allowed a deeper analysis of diabetic skin, which could assist in the evaluation of cosmetic treatments to improve skin alterations caused by the glycation process.

Finally, this study highlights differences between diabetic and healthy skin, making it a promising target for new cosmetic products.

## Figures and Tables

**Figure 1 life-13-00579-f001:**
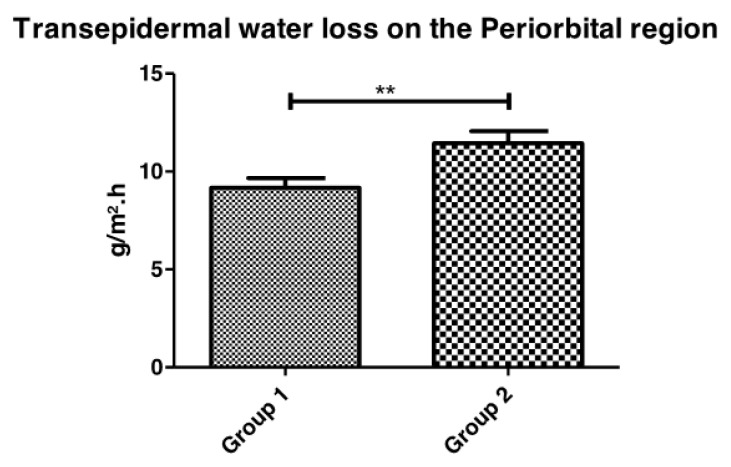
Transepidermal water loss comparison between Group 1 (healthy) and Group 2 (iabetic) participants in the Periorbital region (** *p* = 0.009).

**Figure 2 life-13-00579-f002:**
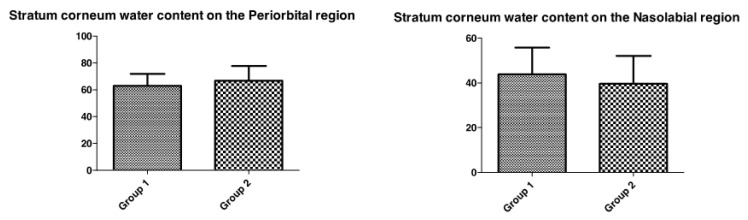
Stratum corneum water content comparison between Group 1 (healthy) and Group 2 (diabetic) participants in Nasolabial and Periorbital region.

**Figure 3 life-13-00579-f003:**
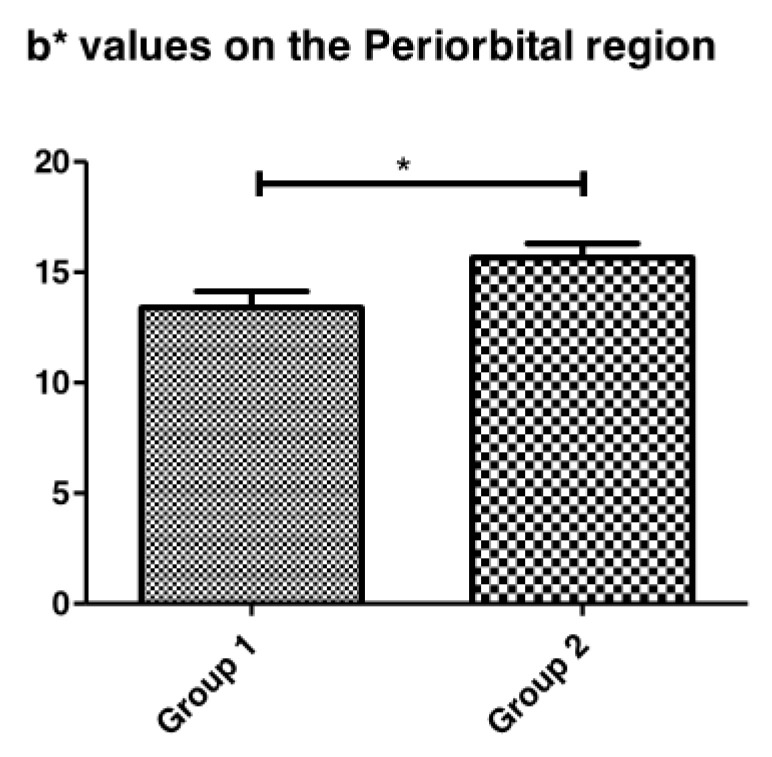
b* values comparison of skin color between Group 1 (healthy) and Group 2 (diabetic) participants in the Periorbital region (* *p* = 0.03).

**Figure 4 life-13-00579-f004:**
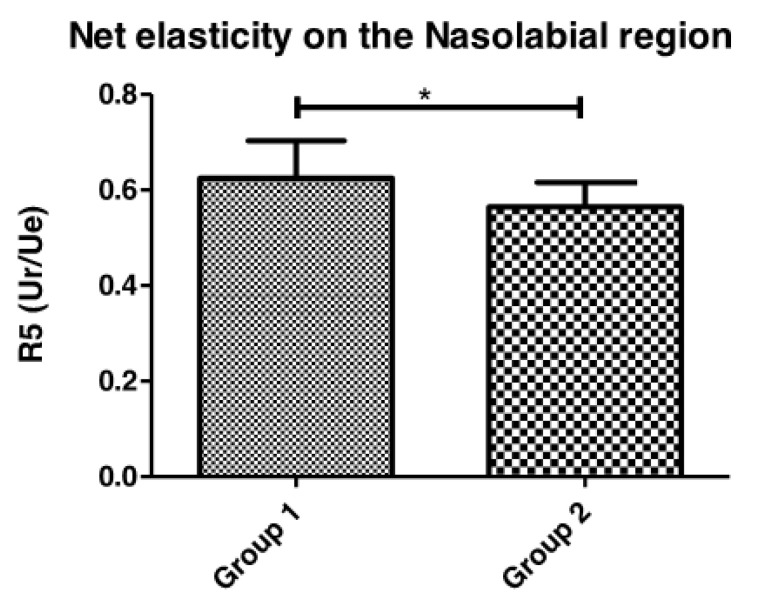
Net elasticity parameter (Ur/Ue) comparison between Group 1 (healthy) and Group 2 (diabetic) participants in the Nasolabial region (* *p* = 0.03).

**Figure 5 life-13-00579-f005:**
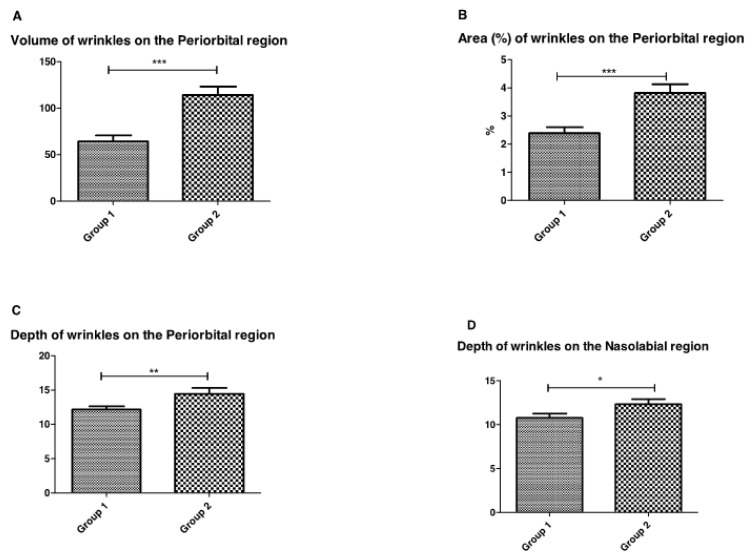
Volume, area, and depth of wrinkles (**A**–**C**) between Group 1 (healthy) and Group 2 (diabetic) participants in the Periorbital region and depth of wrinkles in the Nasolabial region (**D**) (* *p* = 0.02; ** *p* = 0.007, *** *p* = 0.0006).

**Figure 6 life-13-00579-f006:**
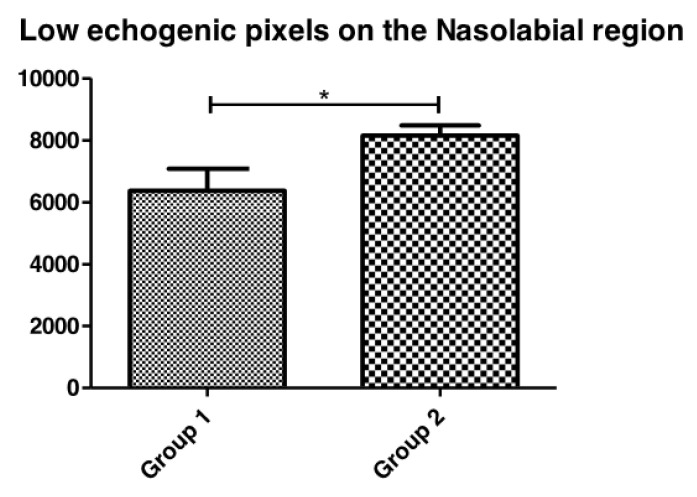
Comparison between low echogenicity pixels of Group 1 (healthy) and Group 2 (diabetic) in the Periorbital region (* *p* = 0.03). Higher values of low echogenicity pixels show more damage in the dermis.

**Figure 7 life-13-00579-f007:**
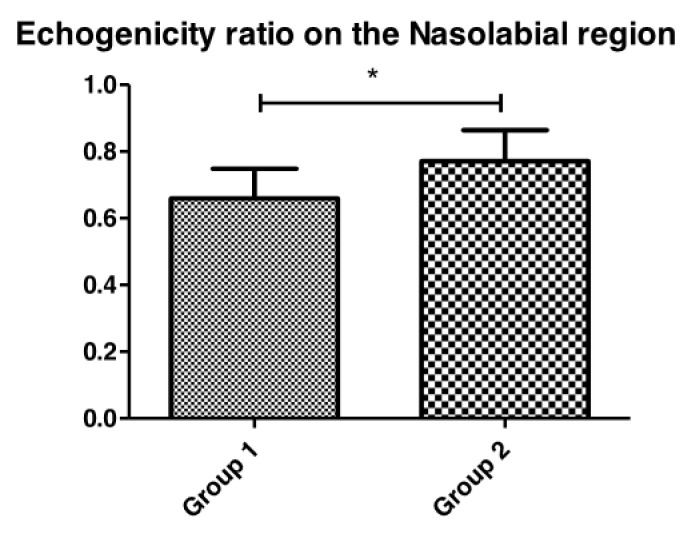
Echogenicity ratio (number of low echogenicity pixels and number of total echogenic pixels—LEP/TEP) on the nasolabial region in the Group 1 (healthy) and Group 2 (diabetic) skin (* *p* < 0.05).

**Figure 8 life-13-00579-f008:**
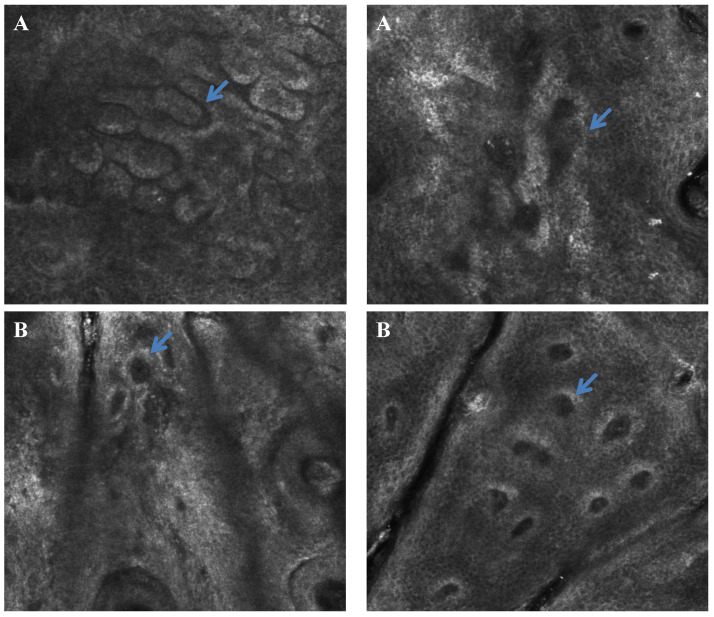
Representative RCM images of dermal–epidermal junction pattern in Group 2 (diabetic) (**A1**,**A2**) and Group 1 (healthy) (**B1**,**B2**) participants. In the diabetic skin, the arrows show deeper multiple polycyclic papillae than in the healthy skin.

**Figure 9 life-13-00579-f009:**
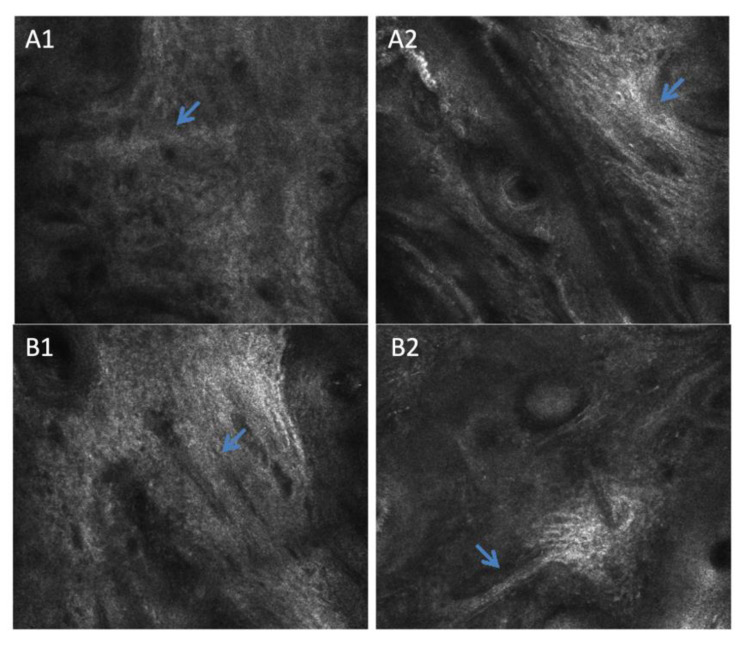
Representative RCM images—The arrows indicate coarse and curled collagen fibers in Group 2 (diabetic) skin (**A1**,**A2**) when compared with Group 1 (healthy), which presented thin reticulated collagen (**B1**,**B2**).

## Data Availability

The data can be shared upon request.

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
