# Peer review of "Evaluation of Morphological and Structural Skin Alterations on Diabetic Subjects by Biophysical and Imaging Techniques"

_life, 2023, doi:10.3390/life13020579_

Round 1
Reviewer 1 Report
Major comments:
- What the authors mean in his part od Introduction: “The research minimizes the chances of having an adverse reaction [9], at the same time helping in the adhesion and loyalty of the consumer with the cosmetic industry that developed it.” Purpose was of their research was to help industry? Authors declared no conflicts of interest.
- Statistical analysis – please describe tests used for checking variables distribution as the authors used parametric and non-parametric tests
- When describing methods, the authors cited own research, but on the other hand described methods in quite detailed manner. So, what is the reason for self-citations? If they consider that description in this paper is adequate to secure reproducibility of the experiments, the citations is not needed. If they want to reduce volume of manuscript, they shall use “as described in…” and use citation. In present form, inappropriate self-citations by authors shall be considered.
- Paper need significant English language improvement
- I don’t think that manuscript falls into journal scope. Because a rate it as interested for particular readers, it shall be published in journal focused on dermatology or cosmetology, but not in general scientific journal.
Minor comments:
- position 26 of references was divided into 26 and 27
- Line 174 “Stratum” shall be written with “s”
- plain filling (e.g. white and black) of the columns in the figures increase readability
Author Response
Reviewer 1:
1- What the authors mean in this part of the Introduction: “The research minimizes the chances of having an adverse reaction [9], at the same time helping in the adhesion and loyalty of the consumer with the cosmetic industry that developed it.” The purpose of their research was to help industry? The authors declared no conflicts of interest.
The research is not intended to help the industry. Because it is a very specific niche and with already fragile skin, the concern of the study was to develop a formulation that would not be able to trigger any type of allergy, as well as a pleasant sensory experience due to the country's climate so that the participants could have a high adherence to the use of the product to avoid unreliable results.
2- Statistical analysis – please describe tests used for checking variables distribution as the authors used parametric and non-parametric tests.
Statistical analysis was performed using Graphpad Prism 5. To determine significant differences in skin characteristics for regions between populations, Student’s t-test was used for parametric data, and the Mann- Whitney U- test was used for non-parametric data. A p-value <0.05 was considered significant.
3- When describing methods, the authors cited own research, but on the other hand described methods in quite detailed manner. So, what is the reason for self-citations? If they consider that description in this paper is adequate to secure reproducibility of the experiments, the citations is not needed. If they want to reduce volume of manuscript, they shall use “as described in…” and use citation. In present form, inappropriate self-citations by authors shall be considered.
Thank you for the comment. The reviewer is right; we removed the not needed self-citations from the manuscript in the methods section.
4- Position 26 of references was divided into 26 and 27.
Modified in the manuscript.
5- Line 174 “Stratum” shall be written with “s”.
Modified in the manuscript.
Reviewer 2 Report
This is an interesting article for evaluation of morphological and structural skin alterations on diabetic subjects by biophysical and imaging techniques. To reach a publishable level, it is critical to improve the following points:
Major criticism
1. Line 51: AGEs can be found in several types of tissues during aging. I suggest to authors to clarify this sentence and to use more citations.
2. Lines 102-104: Provide more information for the participants with diabetes. Did the authors include people with type 1 or type 2 diabetes? Moreover, as glycemic control affect skin characteristics, how was diabetes control of the participants with diabetes? The authors should add a table with demographic and clinical characteristics of the participants.
3. Were there signifivant associations between the studies parameters with age, duration of diabetes or diabetes control? Were there differences between men and women?
4. Line 295-297: Furthermore, some studies demonstrate that this modification in vimentin results in alterations as loss of fibroblast contractility, which could be a factor that accelerates the aging process. I suggest to authors to explain this information in more detail. They should explain the role of vimentin and its correlation with fibroblast contractility and aging.
5. References should be formatted as journal style.
Minor criticism
1. Review the language (grammar, punctuation, and word spelling); Thus, I suggest the authors get editing help from someone with full professional/editorial proficiency in English
2. Line 8: ‘final glycation products (AGEs)’. This definition is not correct; AGEs are advanced glycation end products.
3. Line 12: ‘TEWL’ Authors should define this abbreviation.
4. Line 18: I suggest to authors to replace word important with significant.
5. Line 29: Skin aging without S capital the word skin.
6. Line 33 ‘reations’. The correct word is reactions.
7. Line 43: phrase ‘advanced glycated end products (AGEs)’ is repeated. Authors should define abbreviations and acronyms in the ‘Abstract’ and at first mention them in the main text, and thereafter use the abbreviation only.
8. Abstract and text: instead of using the terms “diabetic” and “diabetic patients” the authors should use the term “people with diabetes”.
9. Line 56: endogenous word does not need capital E.
10. Line 328: noninvasive needs a ‘-’ non-invasive.
11. In the entire text the word “diabetes” is written as” diabetes” and “Diabetes”. The authors should use the term “diabetes”.
Author Response
Reviewer 2:
1- Line 51: AGEs can be found in several types of tissues during aging. I suggest to authors to clarify this sentence and to use more citations.
The excessive accumulation can be motivated by pathologic diseases such as diabetes mellitus, and cardiovascular disease. As humans age, the accumulation of AGEs causes a stiffening of the tissues like tendons, joints, bone, skin, and arteries which can lead to several health problems such as neurodegeneration and diabetic neuropathy (Papachristou, S.; Pafili, K.; Papanas, N. Skin AGEs, and diabetic neuropathy. BMC Endocr Disord. 2021. Richard, D.; Semba, E.J.; Nicklett, L. Does Accumulation of Advanced Glycation End Products Contribute to the Aging Phenotype? The Journals of Gerontology. 2010, 9, 963–975.)
2- Lines 102-104: Provide more information for the participants with diabetes. Did the authors include people with type 1 or type 2 diabetes? Moreover, as glycemic control affect skin characteristics, how was diabetes control of the participants with diabetes? The authors should add a table with demographic and clinical characteristics of the participants.
The study used only participants with type 2 diabetes who were under glycemic control through oral medications such as metformin and medical follow-up for at least 5 years. As they were volunteers from a lower social class, it was not possible to request blood tests to prove glycemic control.
3- Were there significant associations between the studies parameters with age, duration of diabetes or diabetes control? Were there differences between men and women?
To avoid gender bias, only female diabetic participants were used. The longer the disease, the more visible the signs of aging are since glycation occurs longer. Wrinkles and wrinkle depth, especially in the periorbital region, are more prominent in participants with longer disease duration.
4- Line 295-297: Furthermore, some studies demonstrate that this modification in vimentin results in alterations as loss of fibroblast contractility, which could be a factor that accelerates the aging process. I suggest to authors to explain this information in more detail. They should explain the role of vimentin and its correlation with fibroblast contractility and aging.
Lower values in the dermis echogenicity can be correlated with an aged dermis and loss in the dermal structure, including collagen fibers [10]. This restructuring is significant for the tissue repair and organization of the extracellular matrix. Vimentin is essential for various cell functions such as cell motility and wound healing and is the principal target for AGE modification by N-(carboxymethyl)lysine (CML). The vimentin glycation caused by CML originates a modification structural of the filament, which accelerates the process of aging. Furthermore, some studies demonstrate that this modification in vimentin results in alterations such as loss of fibroblast contractility, which could be a factor that accelerates the aging process. This reduction contractile results in a reduction in the ability to in reorganize collagen fibers and rearrange extracellular matrix, which is fundamental for tissue development and repair. In people with diabetes, the difficulty of healing can also be related to glycation since it has an important role in wound healing, and this glycation generates a lower capacity for collagen functionality [25, 26].
5- References should be formatted as journal style.
Done
6- Review the language (grammar, punctuation, and word spelling); Thus, I suggest the authors get editing help from someone with full professional/editorial proficiency in English.
Ok.
7- Line 8: ‘final glycation products (AGEs)’. This definition is not correct; AGEs are advanced glycation end products.
Modified in the manuscript.
8- Line 12: ‘TEWL’ Authors should define this abbreviation.
Added in the manuscript.
9- Line 18: I suggest to authors to replace word important with significant.
Modified in the manuscript.
10-Line 29: Skin aging without S capital the word skin.
Modified in the manuscript.
11- Line 33 ‘reations’. The correct word is reactions.
Modified in the manuscript.
12- Line 43: phrase ‘advanced glycated end products (AGEs)’ is repeated. Authors should define abbreviations and acronyms in the ‘Abstract’ and at first mention them in the main text, and thereafter use the abbreviation only.
Modified in the manuscript.
13-Abstract and text: instead of using the terms “diabetic” and “diabetic patients” the authors should use the term “people with diabetes”.
Modified in the manuscript, however we keep the term diabetic when referring to the skin.
14- Line 56: endogenous word does not need capital E.
Modified in the manuscript.
15- Line 328: noninvasive needs a ‘-’ non-invasive.
Modified in the manuscript.
16- In the entire text the word “diabetes” is written as” diabetes” and “Diabetes”. The authors should use the term “diabetes”.
Modified in the manuscript.
Round 2
Reviewer 1 Report
The quality of mauscript was improved.
Author Response
Thank you for your important comments and suggestions for the improvement of the manuscript. The minor revision of English language was done, as requested.
Reviewer 2 Report
None
Author Response

(The authors gave the same response as above.)
